# Adaptive Query Selection for Camouflaged Instance Segmentation

## ABSTRACT

Camouflaged instance segmentation is a challenging task due to the various aspects such as color, structure, lighting, etc., of object instances embedded in complex backgrounds. Although the current DETR-based scheme simplifies the pipeline, it suffers from a large number of object queries, leading to many false positive instances. To address this issue, we propose an adaptive query selection mechanism. Our research reveals that a large number of redundant queries scatter the extracted features of the camouflaged instances. To remove these redundant queries with weak correlation, we evaluate the importance of the object query from the perspectives of information entropy and volatility. Moreover, we observed that occlusion and overlapping instances significantly impact the accuracy of the selection mechanism. Therefore, we design a boundary location embedding mechanism that incorporates fake instance boundaries to obtain better location information for more accurate query instance matching. We conducted extensive experiments on two challenging camouflaged instance segmentation datasets, namely COD10K and NC4K, and demonstrated the effectiveness of our proposed model. Compared with the OSFormer, our model significantly improves the performance by 3.8% AP and 5.6% AP with less computational cost, achieving the state-of-the-art of 44.8 AP and 48.1 AP with ResNet-50 on the COD10K and NC4K test-dev sets, respectively.

## CCS CONCEPTS

• **Do Not Use This Code → Generate the Correct Terms for Your Paper**; *Generate the Correct Terms for Your Paper*; Generate the Correct Terms for Your Paper; Generate the Correct Terms for Your Paper.

## KEYWORDS

Camouflaged instance segmentation Adaptive query selection Transformer

## 1 INTRODUCTION

Camouflage, originally developed in biology, aims to deceive and confuse prey and predators by using certain concealed methods [48]. Camouflaged objects are adept at utilizing their own structure, lighting, color, and surrounding environment to perfectly imitate other objects in their vicinity [45]. Due to their ability to blend seamlessly

Permission to make digital or hard copies of all or part of this work for personal or classroom use is granted without fee provided that copies are not made or distributed for profit or commercial advantage and that copies bear this notice and the full citation on the first page. Copyrights for components of this work owned by others than the author(s) must be honored. Abstracting with credit is permitted. To copy otherwise, or republish, to post on servers or to redistribute to lists, requires prior specific permission and/or a fee. Request permissions from permissions@acm.org.

*ACM MM, 2024, Melbourne, Australia*

© 2024 Copyright held by the owner/author(s). Publication rights licensed to ACM.

ACM ISBN 978-x-xxxx-xxxx-x/YY/MM

https://doi.org/10.1145/nnnnnnn.nnnnnnn

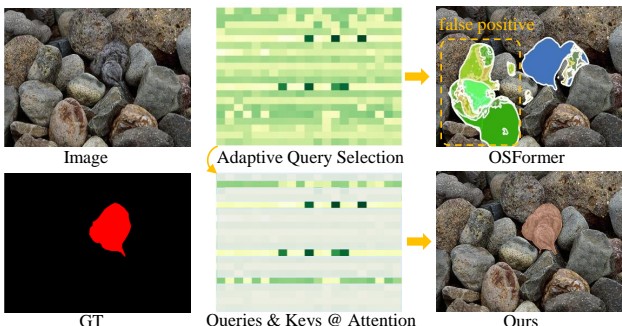

**Figure 1: The core insight of our idea. Current query-based architectures commonly use a large number of queries to collect instance semantic information from the training dataset. However, only a small number of valid queries correspond to object instances, and a large number of redundant in a single image, invalid queries are easy to cause false positive instances. Thus, we propose to promote interactions between valid queries and extracted features while avoiding interactions between useful feature-related information and invalid queries. This approach helps to ensure that the model focuses on the most valid queries, reducing the occurrence of false positive instances.**

into their surroundings, they are difficult to detect. Camouflaged instance segmentation (CIS) [28, 43] is an essential component of camouflaged object analysis, and it is more challenging than general instance segmentation tasks[1, 8, 18, 20, 50]. The establishment of CIS systems in computer-aided intervention perception systems has broad applications in medical diagnosis [13], agriculture [10], security and surveillance, art, and scientific research.

Unlike the two-stage approach used in the first CIS model [28], OSFormer [43] introduces a one-stage algorithm using the query paradigm [3, 19, 35, 41]. This approach employs a concise camouflaged instance segmentation pipeline where each query consists of two functions: content cluster and positional embedding. The query paradigm assumes that each query corresponds to an object instance, and a set of object queries are predefined to map the content and positional information of each query to the corresponding object instance. However, due to the one-to-one correspondence between queries and camouflaged instances, only a few queries are valid, and a large number of queries are redundant, leading to false positive instances and increasing computational costs. While reducing the number of queries may seem intuitive, it has been shown in previous research works that directly reducing the number of queries significantly reduces the model's ability [6, 29]. To address this, we examine the correlation between each query and feature in the cross-attention module. We observe that only a few queries

in the cross-attention are strongly correlated with certain features in the image, while a large number of redundant queries distract the features of the fake instances.

As shown in Figure 1, our proposed approach aims to address the issues of redundant queries and false positive instances. From the perspective of data compression, we want to ensure that after removing redundant queries, we still have the same attention distribution. We find that valid queries exhibit a strong correlation with instance features, while invalid queries show a weak correlation and uniform distribution. To quantify this difference, we use relative entropy to measure the distance between the two distributions and combine it with variance to evaluate query volatility. Instead of using a fixed selection strategy, we propose a multi-head dynamic selection strategy that accommodates dynamic numbers of instances in various scenes. We also observe that overlapping and occluded instances pose challenges for accurate query selection, especially when the positional embedding of the query adopts a random initialization method. To address this issue, we propose using dynamic positional embedding conditioned on the boundary locations of the query input, which allows for more accurate query-instance matching.

By combining the proposed components, we present a novel Adaptive Query Selection Transformer (AQSFormer), which offers an improved end-to-end camouflaged instance segmentation system with adaptive query selection and boundary positional embedding. Our model outperforms the OSFormer and UQFormer by achieving 44.9 AP and 48.1 AP on the COD10K and NC4K test-dev sets with ResNet-50, respectively. This indicates a significant increase of 3.8% AP and 5.6% AP over the previous OSFormer.

## 2 RELATED WORKS

### 2.1 Camouflaged Instance Segmentation

Camouflage is a survival skill resulting from adaptation and natural selection during biological evolution [11]. As a result, biologists have explored numerous examples of camouflage and explained the principles behind it. Inspired by this phenomenon, camouflaged segmentation has become an important research topic in the computer vision community [17, 46, 52, 58], particularly in the context of camouflaged instance segmentation (CIS). Fan *et al.*[16, 17] conducted a comprehensive study of camouflage and published the COD10K dataset, which contains 10K images in 69 categories with rich semantic and instance annotations. To address the challenge of parsing complex scenes with camouflaged objects, researchers have developed various approaches, such as graph convolution[55], uncertainty-guided methods [53], texture difference modeling [31], implicit motion handling [9], receptive field methods [14], and zoom-in and zoom-out methods [42]. However, these approaches only focus on foreground-background separation and cannot distinguish between camouflaged instances.

Recently, Le *et al.*[28] proposed the camouflaged instance segmentation task and designed a two-stage scheme network to address it. Subsequently, Pei*et al.*[43] proposed the first one-stage CIS architecture, which greatly advanced the field of CIS[28, 43]. However, accurate CIS is still a daunting task that faces three major challenges: **(i)** Camouflaged objects tend to change their appearance to perfectly blend into their surroundings, making it challenging

to identify them accurately. **(ii)** Camouflaged objects have various appearances, such as size and shape, which reduce the robustness of the CIS model. **(iii)** Wild animals often live in complex natural environments, which means that images often have complex backgrounds, further exacerbating the difficulty of CIS.

### 2.2 General Query Paradigm

The query paradigm [7, 19, 35, 41] originated from object detection and has evolved into a concise object detection pipeline. The query serves as a learnable external vector, independent of the current input image's content. It aggregates the features of specific object instances in the sequence output from the encoder through cross-attention, models the relationship between object queries using self-attention pairs, and finally, the feed-forward network (FFN) regresses classification detection boxes according to the object query after feature aggregation. The query paradigm has demonstrated strong performance not only in object detection but also in other domains such as panoptic segmentation [6, 54], instance segmentation [8, 18, 43], semantic segmentation [7, 56], crowd counting [33], text detection [38], and human-object interaction detection [25, 26]. Our main structure also builds upon the query paradigm, on which we propose an adaptive query selection strategy for CIS.

### 2.3 Selection Strategy in Transformer

The transformer architecture employs selection strategies [32, 36, 40, 44, 49] to achieve model compression and acceleration by retaining valuable information and removing redundant content. Dynamic ViT [44] uses unstructured sparse hierarchical pruning to dynamically filter tokens for the next layer based on the scores of additional prediction modules. Evit [32] determines the importance of other tokens for the classification task using the class token, and retains valuable tokens based on a fixed ratio while discarding the rest via simple fusion. KVT [49] selects the most similar tokens through KNN clustering to calculate self-attention, thereby removing irrelevant tokens and speeding up training and inference. Adaptive Sparse ViT [36] proposes a minimal-cost adaptive sparse token sparse architecture that discriminates the importance of tokens using a learnable threshold and an inexpensive multi-head attention importance weighted evaluation mechanism. Evo-ViT [51] leverages the global attention advantage in transformers for unstructured instantiated token selection combined with path update and designs a self-motivated slow-fast token evolution method. However, these selection strategies have two limitations: **(i)** They primarily aim to reduce tokens in self-attention to lower computational overhead, which is not suitable for cross-attention with an invalid query. **(ii)** Unifying scenes and objects in classification tasks reduces the challenge of token selection. Applying these strategies directly to complex dense prediction tasks, particularly in CIS, is difficult.

## 3 METHOD

**Problem Statement.** In this study, we consider the feature $X$, which is extracted from the backbone or enhanced by transformer encoder, and a set of object queries $Q = \{q_1, q_2, ..., q_n\}$. Our goal is to learn queries that encode the information of the extracted feature set $X$ into $Q$.

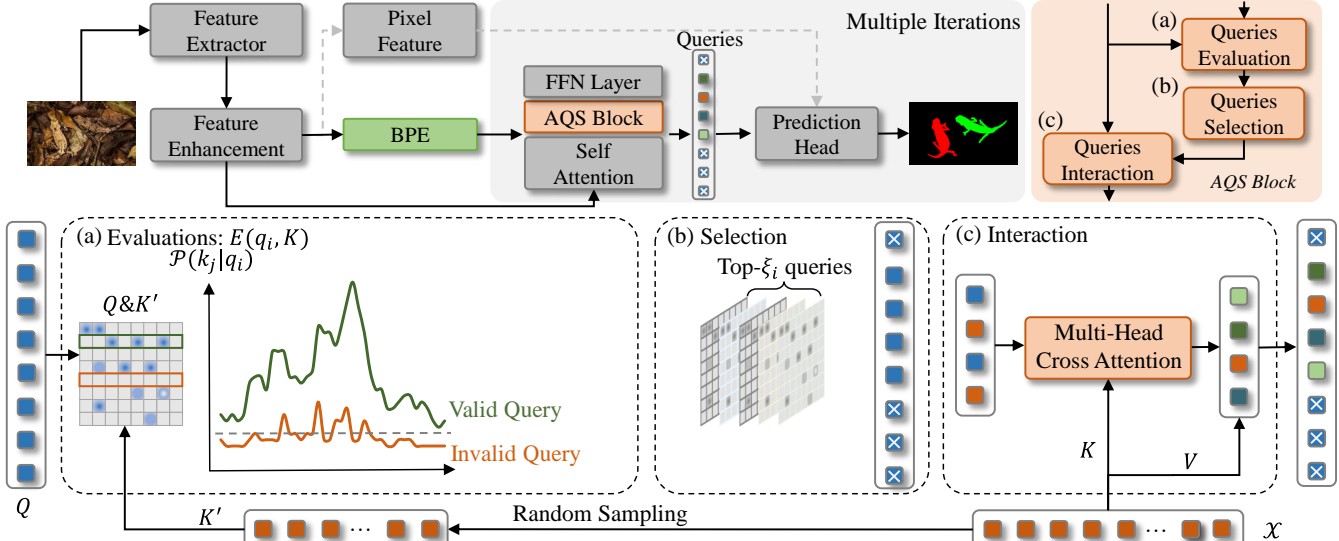

Figure 2: The overall architecture of adaptive query selection. The overall strcture is shown in the figure above, including two core modules, BPE and AQSFormer block. At the same time, we show three key steps in AQSFormer in detail. Figure (a) depicts the process of evaluating the importance of queries, which involves comparing the attention scores obtained with a predefined threshold. Queries with scores above the threshold are marked as valid, while those below it are deactivated. Figure (b) shows the Top-$\xi$ queries selection mechanism operating across multiple heads. In this case, the Top-$\xi$ valid queries from each head are selected based on their importance scores, and the resulting set of queries is used for subsequent computations. Finally, Figure (c) illustrates the interaction between the valid queries and the features in the network. Specifically, the valid queries attend to relevant features, which are then aggregated and processed by the feed-forward network to produce the final output.

**General Cross Attention.** The conventional approach for information mapping involves implementing a cross-attention module between all queries and features. Here, we introduce the positional embedding of the query as $Q_p$ and the content part as $Q_c$. After applying a linear layer, $\mathcal{X}$ is uniformly mapped to a $d$-dimensional representation as key $K$ and value $V$. The query content $Q_c$ and query positional embedding $Q_p$ correspond to the content $K_c$ and positional embedding $K_p$ of $K$, respectively. The attention of the $i$-th query is defined as the probabilistic form of kernel smoothing:

$$\mathcal{A}(q_i, K, V) = \sum_j \frac{k(q_i, k_j)}{\sum_l k(q_i, k_j)} v_j = \mathbb{E}_{\mathcal{P}(k_j|q_i)}[v_j], \quad (1)$$

where $q_i$, $k_i$, and $v_i$ represent the $i$-th row of $Q$, $K$, and $V$, respectively. Here, the attention of the $i$-th query to all keys is defined as the probability $\mathcal{P}(k_j|q_i)$:

$$\mathcal{P}(k_j|q_i) = \frac{k(q_i, k_j)}{\sum_l k(q_i, k_l)}, k(q_i, k_j) = \exp(\frac{q_i k_j^T}{\sqrt{d}}) \quad (2)$$

While general cross attention has been successful in evaluating the correlation between each query and content features with scaled dot-product attention, it also has drawbacks. Firstly, the number of instances in the image is typically much smaller than the number of queries, indicating that most queries are redundant. Retaining all queries is highly prone to false positive instances of these redundant queries, which is especially problematic in camouflaged instance segmentation due to the high similarity between background and

foreground instances. Through visual analysis of $\mathcal{P}(k_j|q_i)$ in Figure 1, we further find that not every retrieved query is valuable, and a large number of queries do not pay special attention to any area during the interaction. Secondly, unlike query $Q$ and key $K$ in self-attention [15] that come from the same feature space and own natural position correspondence, $K$ and $Q$ lack position correlation, making it difficult to accurately correspond.

The natural question that arises *is whether there exists an approach to circumvent the current limitations of the query paradigm.* In response, we propose an adaptive query selection mechanism that provides an affirmative answer.

### 3.1 Valid Query Selection

**Importance evaluations of the queries.** The dot-product attention mechanism is a powerful way for selecting relevant information from extracted features. Its attention probability distribution is key to identifying the most important elements in the input. However, in practice, only a small number of queries are effective, while a large number are redundant and can lead to false positive predictions. Our goal is to select effective queries while retaining the maximum amount of attention information.

From the data compression perspective, we aim to maximize the amount of information after query selection to represent the original attention distribution. Entropy and variance are two key indicators of information quantity. Maximizing entropy and variance is necessary to maximize the amount of information quantity. We use relative entropy with the uniform distribution to quantify entropy, which measures the distance between two distributions

and provides insights into the information quantity gained using the attention mechanism. Variance provides an indication of how well attention is focused on the most important elements. By using these two metrics, we are able to evaluate the effectiveness of our attention mechanism and ensure that it selects the most important information from the input. Ultimately, our approach helps to address the issue of redundant queries in the dot-product attention mechanism and enables more accurate predictions.

As previously mentioned, the difference between the probability distributions $\mathcal{P}(k_j|q_i)$ and the uniform distribution $\mathcal{Q}(k_j|q_i)$ can be used to differentiate valid queries. This difference can be measured using the relative entropy, given by:

$$D_{KL}(\mathcal{P}||\mathcal{Q}) = \sum_{j=0}^{L_k} \mathcal{P}(k_j|q_i)(\log \mathcal{P}(k_j|q_i) - (\log \mathcal{Q}(k_j|q_i)). \quad (3)$$

Here, the uniform distribution is defined as $\mathcal{Q}(k_j|q_i) = \frac{1}{L_k}$, which assumes that the probability of a query attending to each $k_j$ is the same. Using $e$ as the base of the logarithm, the expression for $D_{KL}(\mathcal{P}||\mathcal{Q})$ can be simplified to:

$$D_{KL}(\mathcal{P}||\mathcal{Q}) = \ln \sum_{j=1}^{L_k} \exp(\frac{q_i k_j^T}{\sqrt{d}}) - \frac{1}{L_k} \sum_{j=1}^{L_k} \frac{q_i k_j^T}{\sqrt{d}} - \ln L_k, \quad (4)$$

where the first term is a Log-Sum-Exp (LSE) over all keys, the second term is their arithmetic mean, and the third term is a constant that can be ignored during comparison. We integrate the variable into $M(q_i, K)$, which is an approximation of the evaluations of the queries expressed as the difference between the maximum and the mean. However, the LSE computation may lead to numerical stability issues, such as overflow or underflow. To address this, we approximate LSE using the convex property of the function, resulting in the following approximation for $M(q_i, K)$:

$$M(q_i, K) = \max_j(\frac{q_i k_j^T}{\sqrt{d}}) - \frac{1}{L_K} \sum_{j=1}^{L_k} \frac{q_i k_j^T}{\sqrt{d}}. \quad (5)$$

It can be found that an approximation of the evaluations of the queries can be expressed as the difference between the maximum and the mean. In order to avoid the influence of noise, we introduce the variance of the $q(k_j|q_i)$ with scale factor $\lambda$ to further estimate the degree of dispersion, so the final importance evaluation measure $E(q_i, K)$ can be expressed as:

$$E(q_i, K) = M(q_i, K) + \lambda \sigma^2(\frac{q_i K^T}{\sqrt{d}}). \quad (6)$$

However, computing the importance score of the queries over all keys brings additional computation, so we only compute a fraction of the keys sampled under the assumption that the dot product results follow a long-tailed distribution. Therefore, we can calculate $M(q_i, K)$ by randomly sampling the set of $L_q \ln L_k$ dot product pairs. The $L_k$ and $L_q$ represent the length of keys and queries.

**Adaptive Top-$\xi$ queries selection.** The Top-$\xi$ strategies used in scale-dot production in transformers [49] suffer from a key drawback: the fixed value of $\xi$. This is problematic because there can be significant variations in the number of instances contained in each sample image, rendering a fixed $\xi$ meaningless for a single

sample. To address this issue, we propose a method for dynamically selecting an appropriate number of queries for each sample.

Specifically, we first count the most typical queries in each head which is more importance than the average in the current head. We then calculate the mean of the selected query numbers of all heads as the selected query number of the image. To facilitate mini-batch training , the largest $\xi$ in the mini-batch sample is chosen as the final $\xi$. Note that this step is not necessary for testing since there is only one sample. The proposed adaptive Top-$\xi$ query selection method overcomes the limitations of fixed $\xi$ values and allows for more appropriate query selection in the context of varying instances across samples.

**Queries Interaction with Features** Ideally, we want valid queries to collect as much instance information as possible, while ignoring invalid queries. To achieve this, we only perform cross-attention on selected valid queries, while preventing invalid queries from interacting with features to avoid introducing false positive instances. We adopt the following strategies:

$$\mathcal{A}(q_i, K, V) = \begin{cases} \mathbb{E}_{P(k_j|q_i)}[v_j], & q_i \in \text{Top-}\xi, \\ 0 & q_i \notin \text{Top-}\xi. \end{cases} \quad (7)$$

There are two aspects to consider when using zero padding directly for invalid queries: (i) Directly throwing away invalid queries is not suitable for iterative optimization. (ii) Using feature-related means to establish the relationship between invalid queries and features is somewhat contrary to our original intention.

## 3.2 Boundary Positional Embedding

To address the second question, we propose the boundary positional embedding method for queries. Our approach involves deriving the positional embedding of each query from the important boundary points of the corresponding object instance. We take into account the isotropic nature of the objects in all directions and extract the boundary feature by compensating for the contour using the Laplacian second-order operator, denoted by $\nabla^2(\cdot)$. The boundary feature $H_e$ can be obtained from the feature $X$ as follows:

$$H_e = \nabla^2(f_{\text{med}}(X, ks), ks), \quad (8)$$

where $ks$ is the kernel size and is set to 3. Since the Laplacian operator is a second-order operator, it is more sensitive to noise. Hence, we first use the median blur filter $f_{\text{med}}(\cdot)$ for noise reduction.

We observed that directly selecting the Top-$\kappa$ features can result in a clustering effect, which leads to invalidation of the Top-$\kappa$ feature selection. To avoid this, we filter out the highest local response point in a patch of $s \times s$ as follows:

$$H = \max_{i=1, j=1}^{s,s} H_e(i, j), \quad (9)$$

The boundary information represented by different scale features is more abundant, so we combine the multi-scale features to obtain the final edge features. Finally, we select the most representative boundary feature point denoted as $Q_p$, which is the Top-$\kappa$ feature:

$$Q_p = \text{Top-}\kappa(H), \quad (10)$$

where Top-$\kappa$ is equal to the number of queries.

## 3.3 Network Instantiation

**Feature Extraction and Enhancement.** As shown in Figure 2, the feature extraction of our proposed query paradigm architecture comprises a feature extrctor and feature emhancement. To ensure a fair comparison with previous works, we employ CNNs such as ResNet-50 [22] and ResNet-101 [22] as the extrctor to extract scale features. We also use a transformer encoder with six layers and deformable attention for feature enhancement, similar to [6, 43, 57]. Finally, we obtain multi-scale feature $X_i, i = \{1, 3\}$ for query learning and a single-scale pixel feature $X_4$ for prediction [6, 30].

**Query Learning.** The query learning process is mainly carried out in the transformer decoder stage. To optimize the query, we adopt a supervision strategy after initialization, using the self-attention layer, AQS block, and feed-forward network layer order in the transformer decoder. Additionally, we employ multiple iterations to optimize the query.

**Prediction Head.** To obtain the final location prediction, the optimized queries from the feature enhancement is passed through a MLP layer. Likewise, the final mask prediction is obtained by embedding into the pixel features [6, 7, 43]. We calculate the loss function using binary matching based on the Hungarian matching algorithm's sample allocation strategy. We use the cross-entropy loss function for location prediction and a combination of dice loss and cross-entropy loss for mask prediction to balance the two prediction tasks. The weightage for location prediction and binary mask prediction is set to 0.5 and 5, respectively.

## 4 EXPERIMENTS AND RESULTS

### 4.1 Implementation Details

**Training Details.** We utilize ResNet-50 [22] and ResNet-101 [22] pre-trained on ImageNet-1k [27] as our backbones. The batch size and number of training iterations are set to 16 and 15,000, respectively. We use the current CIS datasets include COCO [34], COD10K[17], and NC4K [39]. To evaluate the model's accuracy, we measure the common instance segmentation evaluation metrics, including AP, $AP_{50}$, and $AP_{75}$.

### 4.2 Comparisons with State-of-the-arts

**Results on COD10K.** Table 1 presents the results of our proposed AQSFormer and other models for camouflaged instance segmentation on the test set of COD10K. Our algorithm overcomes this challenge and outperforms the CIS-specific model OSFormer and UQFormer. Specifically, on the ResNet-50 backbone, compared with the baseline model, our approach achieves 3.4%, 3.5%, and 5.2% improvements in AP, $AP_{50}$ and $AP_{75}$, respectively. Moreover, our architecture achieves better results than OSForemr, with 3.8%, 0.9% and 5.6% improvements in AP, $AP_{50}$ and $AP_{75}$, respectively. When we report the results on the ResNet-101 backbone, the difference between our architecture on ResNet-50 and ResNet-101 is significantly smaller than that of OSForemr and Mask2Foremr, indicating the robustness of our architecture even with lightweight backbones.

**Results on NC4K dataset.** In Table 1, we also evaluate the generalization performance of the models on the NC4K dataset. Our analysis reveals two key findings: (i) the NC4K dataset is comparatively easier than the COD10K dataset, and (ii) the baseline model,

Mask2Former, outperforms the CIS-specific OSFormer on NC4K. However, our proposed architecture exhibits stable generalization performance on NC4K. Specifically, our model achieves an improvement of 3.5% AP over ResNet-50, and a significant improvement of 5.6% AP over OSFormer.

**Importance Evaluation Strategies.** We first analyze the impact of the query importance evaluation strategies on the performance in Table 2. We can observe that using KL divergence for evaluation is relatively more efficient than using variance (+0.7% AP). This is because the variance is single number related while the KL divergence is distribution related, which is more stable. In other words, variance directly judges the distance between the query and its mean, thereby expressing the degree of dispersion, which is related to its own numerical range. The KL divergence is the similarity between the judgment and the uniform distribution, and has nothing to do with the value. The combination of these two strategies achieves an optimal sparsity measure with almost no computational overhead.

**Adaptive Top-$\xi$ Selection.** The adaptive selection Top-$\xi$ scheme is a critical component of our method, and its value has a significant impact on the final predictions. Previous schemes uses a fixed value of $\xi$ to filter out Top-$\xi$ queries with a relatively high degree of dispersion. However, in our experiments, we find that varying $\xi$ led to better results. Specifically, setting $\xi = 10$ works well, likely due to the number of instances in the dataset. Nonetheless, the fixed Top-$\xi$ method does not allow for customization of sparsity for individual samples, making it difficult to find an appropriate value. By contrast, our adaptive Top-$\xi$ selection mechanism can independently screen each sample, resulting in improvements of 1.0% AP, 1.1% $AP_{50}$, and 1.9% $AP_{75}$ compared to the fixed $\xi = 10$ scheme. As a result, our approach achieves state-of-the-art performance.

**Padding of Invalid Queries.** We conduct a comparison of two schemes for padding invalid queries in Table 2: using the mean value of feature $V$ and setting it to zero. Our results indicate that setting the value to zero results in a 1.0% improvement in AP compared to using the mean. This finding suggests that we expect invalid queries to be irrelevant to the data, and setting their mean value may introduce more interference information. Thanks to the three effective component designs discussed earlier, our adaptive query selection scheme achieves significant improvement. Specifically, as shown in Table 4, our scheme improves AP by 2.6%, $AP_{50}$ by 1.8%, and $AP_{75}$ by 3.5% compared to the baseline.

**Boundary Positional Embedding.** We propose a novel method that improves the accuracy of query selection for overlapping and occluded object instances. In Table 3, we demonstrate the impact of three key components: $f_{med}$, lab, and Laplace change. We find that all three components are indispensable, with lab being the most critical component (-1.9% AP decrease when missing). This highlights the importance of Laplace change, a core component that plays a key role in our method. Additionally, the median filter and max also play important roles in suppressing redundant noise and the aggregation of boundary points. Our method utilizes boundary positional embedding, as shown in Table 4, resulting in a significant improvement of 2.3% AP, 1.3% $AP_{50}$, and 3.2% $AP_{75}$ compared to the baseline. Moreover, building upon our adaptive query selection scheme, we achieve a further improvement of 0.9% AP, 0.8% $AP_{50}$, and 1.2% $AP_{75}$.

**Table 1: Quantitative comparison of camouflaged instance segmentation with 14 SOTA methods on the test set of COD10K [17] and NC4K [39]. We report the results on different backbones (*e.g.*, ResNet-50 [22] and ResNet-101 [22]). For computational complexity comparisons (#Params, #FLOPs), all compared models are tested on the backbone of ResNet-50, with FLOPs averaged over 100 samples.**

| Backbone | ResNet-50 [22] | | | | | | ResNet-101 [22] | | | | | | #Params | #FLOPs |
|---|---|---|---|---|---|---|---|---|---|---|---|---|---|---|
| Dataset | COD10K [17] | | | NC4K [39] | | | COD10K [17] | | | NC4K [39] | | | | |
| Metric | AP | $AP_{50}$ | $AP_{75}$ | AP | $AP_{50}$ | $AP_{75}$ | AP | $AP_{50}$ | $AP_{75}$ | AP | $AP_{50}$ | $AP_{75}$ | | |
| **Two stage methods** | | | | | | | | | | | | | | |
| Mask R-CNN [ICCV17] [21] | 25.0 | 55.5 | 20.4 | 27.7 | 58.6 | 22.7 | 28.7 | 60.1 | 25.7 | 36.1 | 68.9 | 33.5 | 43.9M | 186.3G |
| MS R-CNN [CVPR19] [23] | 30.1 | 57.2 | 28.7 | 31.0 | 58.7 | 29.4 | 33.3 | 61.0 | 32.9 | 35.7 | 63.4 | 34.7 | 60.0M | 198.5G |
| Cascade R-CNN [TPAMI19] [2] | 25.3 | 56.1 | 21.3 | 29.5 | 60.8 | 24.8 | 29.5 | 61.0 | 25.9 | 34.6 | 66.3 | 31.5 | 71.7M | 334.1G |
| HTC [CVPR19] [5] | 28.1 | 56.3 | 25.1 | 29.8 | 59.0 | 26.6 | 30.9 | 61.0 | 28.7 | 34.2 | 64.5 | 31.6 | 76.9M | 331.7G |
| BlendMask [CVPR20] [4] | 28.2 | 56.4 | 25.2 | 27.7 | 56.7 | 24.2 | 31.2 | 60.0 | 28.9 | 31.4 | 61.2 | 28.8 | 35.8M | 233.8G |
| Mask Transfiner [CVPR22] [24] | 28.7 | 56.3 | 26.4 | 29.4 | 56.7 | 27.2 | 31.2 | 60.7 | 29.8 | 34.0 | 63.1 | 32.6 | 44.3M | 185.1G |
| **One stage methods** | | | | | | | | | | | | | | |
| YOLACT [ICCV19] [1] | 24.3 | 53.3 | 19.7 | 32.1 | 65.3 | 27.9 | 29.0 | 60.1 | 25.3 | 37.8 | 70.6 | 35.6 | - | - |
| CondInst [ECCV20] [47] | 30.6 | 63.6 | 26.1 | 33.4 | 67.4 | 29.4 | 34.3 | 67.9 | 31.6 | 38.0 | 71.1 | 35.6 | 34.1M | 200.1G |
| QueriesInst [ICCV21] [18] | 28.5 | 60.1 | 23.1 | 33.0 | 66.7 | 29.4 | 32.5 | 65.1 | 28.6 | 38.7 | 72.1 | 37.6 | - | - |
| SOTR [ICCV21] [20] | 27.9 | 58.7 | 24.1 | 29.3 | 61.0 | 25.6 | 32.0 | 63.6 | 29.2 | 34.3 | 65.7 | 32.4 | 63.1M | 476.7G |
| SOLOv2 [NIPS20] [50] | 32.5 | 63.2 | 29.9 | 34.4 | 65.9 | 31.9 | 35.2 | 65.7 | 33.4 | 37.8 | 69.2 | 36.1 | 46.2M | 318.7G |
| SparseInst [CVPR22] [8] | 32.8 | 60.5 | 31.2 | 34.3 | 61.3 | 32.8 | 36.0 | 63.2 | 35.4 | 38.3 | 65.9 | 37.8 | 31.6M | 165.8G |
| Mask2Former [CVPR22] [6] | 41.4 | 68.5 | 41.6 | 44.6 | 71.7 | 45.7 | 44.3 | 70.5 | 46.0 | 49.2 | 76.2 | 51.4 | 44.0M | 232.0G |
| OSFormer [ECCV22] [43] | 41.0 | 71.1 | 40.8 | 42.5 | 72.5 | 42.3 | 42.0 | 71.3 | 42.8 | 44.4 | 73.7 | 45.1 | 46.6M | 324.7G |
| UQFormer [ACMMM23] [12] | **45.2** | 71.6 | **46.6** | 47.2 | 74.2 | 49.2 | 45.4 | 71.8 | 47.9 | 50.1 | **76.8** | 52.8 | 37.5M | 221.0G |
| **AQSFormer** [Ours] | 44.8 | **72.0** | 46.4 | **48.1** | **74.3** | **50.4** | **46.5** | **73.8** | **48.5** | **50.5** | **76.8** | **53.5** | 34.4M | 200.5G |

**Table 2: Ablation analysis of internal components of adaptive query selection.**

| Step | Strategy | AP | $AP_{50}$ | $AP_{75}$ |
|---|---|---|---|---|
| Evaluation | Variance | 46.8 | 73.5 | 48.5 |
| | KL divergence | 47.5 | 74.1 | 50.1 |
| | **Together** [Ours] | **48.1** | **74.3** | **50.4** |
| Selection | Fixed $\xi = 10$ | 47.1 | 73.2 | 48.9 |
| | Fixed $\xi = 15$ | 46.1 | 72.6 | 47.8 |
| | **Dynamic** [Ours] | **48.1** | **74.3** | **50.4** |
| Padding | Mean | 47.1 | **73.4** | 48.9 |
| | **Zero** [Ours] | **48.1** | **74.3** | **50.4** |

**Table 3: Effectiveness of our boundary positional embedding on NC4K dataset. $f_{med}$ and $\nabla^2$ denote the median blur filter, the Laplacian second-order operator, respectively. max is to filter out the highest local response point.**

| $f_{med}$ | $\nabla^2$ | max | AP | $AP_{50}$ | $AP_{75}$ |
|---|---|---|---|---|---|
| | ✓ | ✓ | 46.8 | 73.2 | 49.0 |
| ✓ | | ✓ | 46.2 | 72.5 | 47.8 |
| ✓ | ✓ | | 47.6 | 73.7 | 49.6 |
| ✓ | ✓ | ✓ | **48.1** | **74.3** | **50.4** |

We also demonstrate the effectiveness of boundary extraction in Figure 3. We observe that even without supervision from the ground truth of the boundary, we can obtain clear boundary features. Moreover, the extracted boundaries are not continuous, which aligns with our expectations. Sparse boundary representation can avoid

**Table 4: Comparison of volatility evaluation methods for query on the NC4K dataset. "AQS" and "BPE" are the abbreviation of adaptive query selection and boundary positional embedding, respectively.**

| baseline | AQS | BPE | AP | $AP_{50}$ | $AP_{75}$ |
|---|---|---|---|---|---|
| ✓ | | | 44.6 | 71.7 | 45.7 |
| ✓ | ✓ | | 47.2 | 73.5 | 49.2 |
| ✓ | | ✓ | 46.9 | 73.0 | 48.9 |
| ✓ | ✓ | ✓ | **48.1** | **74.3** | **50.4** |

the aggregation effect caused by sampling Top-$\kappa$ boundary points and has better discriminative ability for camouflaged instances. In Figure 4, we provide additional evidence demonstrating the impact of boundary positional embedding (w/o BPE) on detecting occluded object instances. Our method successfully detects occluded objects with high accuracy.

**Interactive Method Comparisons.** To provide a comprehensive evaluation of our adaptive query selection scheme, we compare it with two widely used interaction modules, namely cross-attention and masked attention. The results are presented in Table 6. Our adaptive query selection scheme outperforms masked attention in terms of improving the removal of redundant query interference. While masked attention primarily focuses on foreground regions, our design maximizes the adaptiveness of query selection, resulting in an improvement of 0.9% AP, 0.7% $AP_{50}$, and 0.8% $AP_{75}$.

**More Transformer Backbones** We find that the backbone of the swin transformer can stimulate the potential of the proposed model

**Table 5: Performance of different transformer backbone.**

| | AP | AP$_{50}$ | AP$_{75}$ |
|---|---|---|---|
| Mask2Former Swin-Tiny | 46.7 | 73.4 | 49.1 |
| OSFormer Swin-Tiny | 46.8 | 73.5 | 49.1 |
| **AQSFormer Swin-Tiny** [Ours] | 54.8 | 80.6 | 58.5 |
| **AQSFormer Swin-Base** [Ours] | 57.9 | 83.5 | 62.0 |
| **AQSFormer PVT-B0** [Ours] | 47.2 | 70.5 | 48.9 |
| **AQSFormer PVT-B1** [Ours] | 50.8 | 74.4 | 53.3 |
| **AQSFormer PVT-B2** [Ours] | 53.2 | 78.5 | 57.1 |

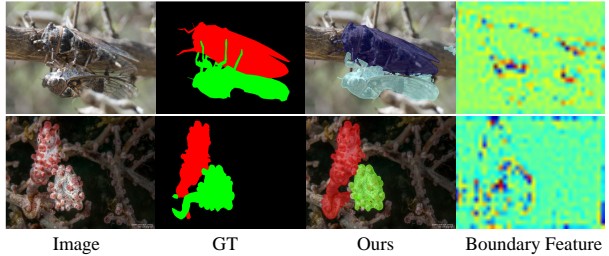

| Image | GT | Ours | Boundary Feature |

**Figure 3: The boundary features extracted in boundary positional embedding**

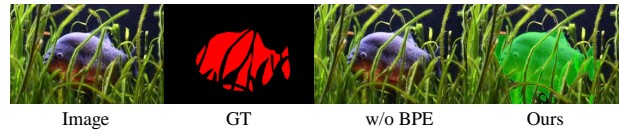

| Image | GT | w/o BPE | Ours |

**Figure 4: Advantages of boundary position encoding.**

**Table 6: Comparisons of interactive methods between feature and queries.**

| Setting | AP | AP$_{50}$ | AP$_{75}$ |
|---|---|---|---|
| Cross attention | 46.9 | 73.0 | 48.9 |
| Masked attention | 47.2 | 73.0 | 49.6 |
| **AQSFormer** [Ours] | **48.1** | **74.3** | **50.4** |

more significantly than the Mask2Former and OSFormer. Therefore, we show more results with different vision transforemr backbone (eg., PVT, Swin). As shown in Table 8, we compare the benefits of two different backbones (PVT, Swin), and find that compared to PVT, Swin performs better in CIS. This is conducive to inspiring subsequent research on the backbone of the transformer.
**Computational Cost.** Furthermore, we compare the computational complexity of these models, and find that our model achieves the best trade-off between accuracy and computational cost, as shown in Table 1. Specifically, the parameters and FLOPs reduce 26.1% and 38.3%, respectively, compared to OSFormer. We also investigate the effect of the number iteration of transformer encoder and decoder, as shown in Table 7. We find that increasing the number of encoder iterations has a more significant impact on performance than increasing the decoder iterations, as more encoder iterations allowed us to extract more discriminative features.

**Table 7: Comparison of different encoder and decoder layers on the NC4K dataset. "AP" refers to the AP results of COD10K/NC4K.**

| Encoder | AP | #Params | #FPS | #FLOPs | #Memory |
|---|---|---|---|---|---|
| Mask2Former | 41.4/44.3 | 44.0M | 17.9 | 230.0G | 8.3G |
| OSFormer | 41.0/42.5 | 46.6M | 23.6 | 324.7G | 6.5G |
| 3 #E, 1 #D | 42.3/44.1 | 29.0M | 27.0 | 155.9G | 4.4G |
| 3 #E, 2 #D | 41.9/45.0 | 30.6M | 25.6 | 156.0G | 4.5G |
| 3 #E, 3 #D | 42.4/45.3 | 32.2M | 25.0 | 156.1G | 4.6G |
| 6 #E, 1 #D | 44.5/47.3 | 31.2M | 23.8 | 200.3G | 6.0G |
| 6 #E, 2 #D | 44.1/46.8 | 32.8M | 22.9 | 200.4G | 6.1G |
| 6 #E, 3 #D | 44.8/48.1 | 34.4M | 21.6 | 200.5G | 6.2G |

**Table 8: Performance under transformer backbone (*e.g.*, Swin tiny transformer [37]).**

| | AP | AP$_{50}$ | AP$_{75}$ | #FLOPs |
|---|---|---|---|---|
| Mask2Former | 46.7 | 73.4 | 49.1 | 238.3G |
| OSFormer | 46.8 | 73.5 | 49.1 | 331.2G |
| **AQSFormer** [Ours] | **54.8** | **80.6** | **58.5** | **206.7G** |

**Table 9: Comparisons on COCO instance segmentation dataset.**

| Setting | AP | #Params | #FLOPs |
|---|---|---|---|
| Mask2former | 43.7 | 44.0M | 230.0G |
| **AQSFormer** [Ours] | **44.5** | **34.4M** | **200.5G** |

**General Instance Segmentation Task.** Moreover, we apply our proposed model to the general instance segmentation task and report competitive experimental results on the COCO instance segmentation task in Table 9. Our approach outperforms Mask2former by 0.8% using less parameters and FLOPs.

## 5 CONCLUSION

In this study, we investigate the issue of query redundancy in camouflage instance segmentation by examining the correspondence between queries and instances, as well as the interaction between queries and features. Our contribution is three-fold: Firstly, we propose an adaptive query selection mechanism based on information entropy and variance, which effectively addresses the issue of redundant queries. Secondly, we introduce a boundary position embedding that incorporates the boundaries of camouflaged instances, which addresses the challenges of occlusion and overlapping instances that affect the query selection. Finally, we conduct extensive experiments on two challenging datasets, demonstrating the superior performance of our model exceeds the cutting edge methods. Additionally, we perform a detailed comparative analysis of each component of our design scheme to demonstrate its effectiveness.

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
