# OpenReview forum: "Adaptive Query Selection for Camouflaged Instance Segmentation"
_acmmm.org/ACMMM/2024/Conference — MM2024 Poster_

### Official Review · Reviewer_Bb98 · 2024-05-26

**Rating:** 5
**Confidence:** 2

**Summary:**

The paper proposed a method for CIS by addressing query redundancy issues via adaptive query selection mechanism and boundary position embedding. The proposed method achieved better performance than existing CIS methods.

**Strengths:**

_ The paper is written well and easy to follow.
_ The proposed method achieved better performance than existing CIS methods.

**Limitations:**

_ Contributions are described as unclear.
_ The authors did not report the processing time.

**Suitability:**

2

---

### Official Review · Reviewer_Ep82 · 2024-05-27

**Rating:** 5
**Confidence:** 3

**Summary:**

The paper addresses the problem of camouflaged instance segmentation (CIS), which involves detecting and segmenting objects that are difficult to distinguish from their backgrounds due to their ability to blend in. The current DETR-based (DEtection TRansformers) methods, while simplifying the segmentation pipeline, suffer from high false positive rates due to the use of a large number of object queries. To mitigate this issue, the authors propose an adaptive query selection mechanism. This mechanism evaluates the importance of object queries using information entropy and volatility, aiming to remove redundant queries and focus on the most relevant ones. Additionally, the paper introduces a boundary location embedding mechanism to improve query-instance matching accuracy by incorporating fake instance boundaries. Extensive experiments on the COD10K and NC4K datasets show that the proposed model outperforms existing methods like OSFormer, achieving state-of-the-art performance with reduced computational cost.

**Strengths:**

•	Novelty:
The introduction of an adaptive query selection mechanism to handle the high false positive rates in camouflaged instance segmentation is a novel contribution. By evaluating the importance of queries based on information entropy and volatility, the method effectively reduces redundant queries.

•	Technical Correctness:
The proposed method is theoretically sound, utilizing well-established concepts such as information entropy, variance, and boundary location embedding. The approach is rigorously justified and systematically developed.

•	Adequate Evaluation:
Extensive experiments on two challenging datasets (COD10K and NC4K) demonstrate the effectiveness of the proposed model. The results show significant performance improvements over the state-of-the-art methods (e.g., OSFormer), with clear metrics provided (3.8% and 5.6% AP improvements).

•	Clarity:
The paper is well-structured and clearly written, with detailed explanations of the proposed mechanisms and thorough discussion of the results. The inclusion of figures to illustrate key concepts and the overall architecture aids in understanding.

•	Applications:
The method has potential applications in various fields such as medical diagnosis, agriculture, security and surveillance, and scientific research, where detecting camouflaged objects is crucial.

**Limitations:**

•	Lack of Novel Datasets:
While the proposed method is tested on established datasets (COD10K and NC4K), the paper could have included new datasets or additional diverse datasets to further validate the robustness of the approach across different scenarios.

•	Limited Comparison with Broader Methods:
The comparison is primarily focused on OSFormer. Including a broader range of baseline methods from related fields, such as other advanced instance segmentation techniques or different transformer-based architectures, could strengthen the evaluation.

•	Complexity of Implementation:
The adaptive query selection mechanism and boundary location embedding introduce additional complexity to the implementation. The paper could benefit from a more detailed discussion on the computational overhead and practical deployment considerations.

•	Occlusion and Overlapping Instances:
While the paper addresses occlusion and overlapping instances with a boundary location embedding mechanism, a more in-depth analysis and discussion on how well this approach handles these challenges compared to other existing methods would be beneficial.

**Suitability:**

3

---

### Official Review · Reviewer_iUC4 · 2024-06-04

**Rating:** 4
**Confidence:** 2

**Summary:**

This paper presents a novel approach to improving camouflaged instance segmentation (CIS) by addressing the issue of redundant queries and false positives in DETR-based schemes.

**Strengths:**

The paper clearly identifies the challenges in camouflaged instance segmentation, such as the difficulty of detecting objects embedded in complex backgrounds due to various factors like color, structure, and lighting. This relevance is well-established by linking it to applications in medical diagnosis, agriculture, security, and more.

The paper provides extensive experimental results on two challenging datasets, COD10K and NC4K, demonstrating significant improvements over existing methods (OSFormer and UQFormer). The detailed comparison and reported improvements in AP metrics highlight the effectiveness of the proposed model.

**Limitations:**

The proposed model, while innovative, adds a layer of complexity that may be challenging to implement and optimize in practice. Further clarity on the computational overhead and efficiency would be beneficial.

Have you considered the robustness of your approach to variations in lighting conditions and object orientations in real-world scenarios?

Extending the evaluation to additional datasets and including more recent state-of-the-art comparisons would strengthen the paper's claims of generalizability and superiority.

**Suitability:**

3

---

### Meta-Review · Area_Chair_u1rP · 2024-06-30

**Recommendation:** Accept (Poster)
**Confidence:** 5

**Metareview:**

This paper has received two WAs, and one BA as initial scores. All reviewers highlight the novelty of the proposed work. They agree that the paper is well-written.

Reviewer iUC4 has concern about the robustness. Reviewers iUC4 and Ep82 question the benchmark datasets. Reviewer Bb98 raises concerns about the processing time.

The authors provide a rebuttal which addresses the reviewers' concerns. The AC agrees with the reviewers that the paper should be accepted.